# Vision-by-Language for Training-Free Compositional Image Retrieval

**Shyamgopal Karthik**[1,*]**, Karsten Roth**[1,*]**, Massimiliano Mancini**[2]**, Zeynep Akata**[1,3,4]

[1]Tübingen AI Center & University of Tübingen, [2]University of Trento
[3]Helmholtz Munich, [4]Technical University of Munich
[*]equal contribution

## ABSTRACT

Given an image and a target modification (e.g an image of the Eiffel tower and the text "without people and at night-time"), Compositional Image Retrieval (CIR) aims to retrieve the relevant target image in a database. While supervised approaches rely on annotating triplets that is costly (i.e. query image, textual modification, and target image), recent research sidesteps this need by using large-scale vision-language models (VLMs), performing Zero-Shot CIR (ZS-CIR). However, state-of-the-art approaches in ZS-CIR still require training task-specific, customized models over large amounts of image-text pairs. In this work, we propose to tackle CIR in a training-free manner via our Compositional Image Retrieval through Vision-by-Language (`CIReVL`), a simple, yet human-understandable and scalable pipeline that effectively recombines large-scale VLMs with large language models (LLMs). By captioning the reference image using a pre-trained generative VLM and asking a LLM to recompose the caption based on the textual target modification for subsequent retrieval via e.g. CLIP, we achieve modular language reasoning. In four ZS-CIR benchmarks, we find competitive, in-part state-of-the-art performance - improving over supervised methods. Moreover, the modularity of `CIReVL` offers simple scalability without re-training, allowing us to both investigate scaling laws and bottlenecks for ZS-CIR while easily scaling up to in parts more than double of previously reported results. Finally, we show that `CIReVL` makes CIR human-understandable by composing image and text in a modular fashion in the language domain, thereby making it intervenable, allowing to post-hoc re-align failure cases. Code available at github.com/ExplainableML/Vision_by_Language.

## 1 INTRODUCTION

Compositional Image Retrieval (CIR) necessitates a nuanced coupling between the image content and the semantics of the textual query to retrieve a new image that accurately embodies the relevant image elements and the modifications described in the textual query. To achieve this, previous works require curated triplets (query image, modifying text, target image) to train a specific CIR system. However, annotating such triplets is both difficult and labor-intensive. To tackle this problem, recent research proposed Zero-Shot CIR (ZS-CIR) (Saito et al., 2023; Baldrati et al., 2023). Based on large-scale pre-trained vision-language models (VLMs) (e.g. CLIP (Radford et al., 2021)), these methods use image-caption pairs to train textual inversions (Gal et al., 2023; Cohen et al., 2022) mapping images to text tokens. A static template merges tokens and textual modifications to obtain target captions, performing CIR without explicit supervision. Thus, even when leveraging large-scale VLMs, ZS-CIR methods still train additional mapping networks on large image-caption datasets.

In this work, we propose to achieve training-free ZS-CIR by leveraging ubiquitously available, off-the-shelf models already trained with large-scale training data. Our Compositional Image Retrieval through Vision-by-Language (`CIReVL`) follows the vision-by-language paradigm (Zeng et al., 2023; Zhu et al., 2023; Berrios et al., 2023; Levy et al., 2023a), which uses language as an abstraction layer for reasoning about visual content. Specifically, `CIReVL` employs vision-language models like BLIP-2 (Li et al., 2023) or CoCa (Yu et al., 2022) to generate a detailed description of the query image. Subsequently, a LLM (s.a. Llama (Touvron et al., 2023) or GPT (Brown et al., 2020))

crafts a caption for the desired target image, using both the generated description and the respective textual query. Finally, a VLM like CLIP (Radford et al., 2021) retrieves the image. `CIReVL` abstracts away the compositional nature of the problem into the language domain, converting ZS-CIR into an inherently modular task comprising captioning, reasoning and cross-modal retrieval.

This opens up various benefits not available in traditional, trained ZS-CIR approaches. Beyond not requiring additional adaptation resources, the training-free and modular nature offers the flexibility for simple model changes and replacements, allowing us to scale up `CIReVL` using freely available models. Consequently, while `CIReVL` already matches and even outperforms trained methods on the common ZS-CIR benchmarks CIRCO (Baldrati et al., 2023) and CIRR (Liu et al., 2021) using comparable model architectures, simple plug-and-play of large retrieval models raises improvements significantly, in parts more than doubling previous results. In addition, `CIReVL` is modular and operates primarily in the language domain, as the outputs of the captioning module and the LLM-generated modifications are textual, offering a degree of understanding over the compositional retrieval process to humans. This is further reflected in the ability for possible human intervention on the retrieval process to fix or post-hoc improve results (Fig.4). Finally, we show the generality of `CIReVL` on domain conversion (Saito et al., 2023) and conditional image similarity (Vaze et al., 2023), and ablation studies elucidate the role of each pipeline component.

Overall, our contributions are: 1) We explore training-free zero-shot compositional image retrieval, proposing `CIReVL`, a new approach that matches or outperforms existing training-based methods on four CIR benchmarks while only relying on off-the-shelf available pre-trained models. 2) We show how the inherent modularity of `CIReVL` and its reasoning over the textual query in the language domain facilitates a degree of human understanding over the compositional retrieval process, even allowing for user-level intervention. 3) We conduct multiple additional studies to ablate pipeline components and point to the importance of language-level reasoning over the textual query, while highlighting the simple scalability of `CIReVL` through its modular, training-free nature.

## 2 RELATED WORK

**Compositional Image Retrieval.** The task of Compositional Image Retrieval has found significant application in conditional search (Wu et al., 2021; Han et al., 2017; Vo et al., 2019), where users perform interactive dialogue to refine a given query image toward retrieving specific items. Classical techniques often employ custom models that project text-image pairs into a common embedding space (Vo et al., 2019; Baldrati et al., 2022; Chen et al., 2020; Chen & Bazzani, 2020; Lee et al., 2021; Anwaar et al., 2021) using contrastive objectives (Sohn, 2016; Radford et al., 2021; Roth et al., 2022b) or cross-modal attention (Delmas et al., 2022) and is closely related to compositional learning (Misra et al., 2017; Mancini et al., 2021; Karthik et al., 2022). With the advent of vision-language foundation models (Bommasani et al., 2021; Radford et al., 2021; Jia et al., 2021), interest in CIR has surged, especially in zero-shot settings without task-specific models. Two prominent directions exist: one using pseudo-tokens to represent reference images, which are then concatenated with the reference caption (Saito et al., 2023; Baldrati et al., 2023; bai et al., 2024); the other trains foundation models on curated triplets tailored for CIR (Liu et al., 2023; Gu et al., 2023; Ventura et al., 2024; Levy et al., 2023b). We explore a different pipeline by coupling VLMs with LLMs to address CIR without any specialized training, in an effective and interpretable manner.

**Vision-Language Models.** Models like CLIP (Radford et al., 2021) and ALIGN (Jia et al., 2021) have been trained on expansive datasets such as LAION-400M/5B (Schuhmann et al., 2021; 2022), enabling them to map images and text into a shared embedding space. These models have seen wide-ranging usage, from generative tasks (Rombach et al., 2022; Ramesh et al., 2022; Gafni et al., 2022; Liu et al., 2022; Chefer et al., 2023; Karthik et al., 2023) to open-vocabulary classification (Radford et al., 2021; Ilharco et al.; Menon & Vondrick, 2023; Pratt et al., 2023; Udandarao et al., 2023; Roth et al., 2023) and (cross-modal) retrieval (Bogolin et al., 2022; Bain et al., 2022; Roth et al., 2022a; Wu et al., 2023). Further advancements include models like BLIP (Li et al., 2022; 2023) and Flava (Singh et al., 2022), which extend beyond shared space projection to address other vision-language tasks like captioning (Vinyals et al., 2016) and visual question answering (Antol et al., 2015). While these models have been indirectly applied to CIR through specialized modules (Vo et al., 2019; Baldrati et al., 2022; Delmas et al., 2022) and with fine-tuning (Gu et al., 2023), our work demonstrates that vision-language models alone, when partnered with an LLM, can suffice for effective CIR without additional training.

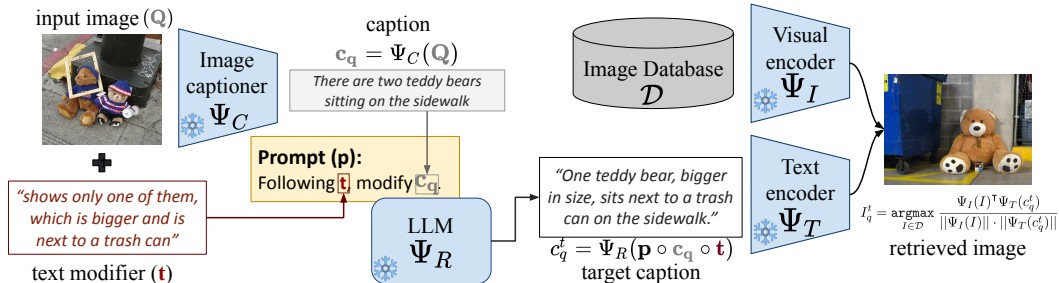

Figure 1: Compositional Image Retrieval through Vision-by-Language (CIReVL). Given an input image and a text modifier, we use an off-the-shelf vision-language model to caption the image. A LLM processes the generated caption and the text modifier to generate a description of the desired target image. To obtain the final image, we use a vision-language model and perform text-to-image retrieval. CIReVL is modular, training-free and human-understandable in natural language.

## 3 CIReVL METHODOLOGY

This section first details the specifics of the ZS-CIR task in §3.1, before presenting our proposed approach, Compositional Image Retrieval through Vision-by-Language (CIReVL), in §3.2.

### 3.1 PRELIMINARIES

Let us define as $\mathcal{I}$ and $\mathcal{T}$ the image and text space, respectively. For compositional image retrieval (CIR), a text modifier $t \in \mathcal{T}$ describes hypothetical semantic changes on a query image $Q \in \mathcal{I}$, whose closest realization $I_q^t \in \mathcal{D}$ from some image database $\mathcal{D} = \{I_i, \cdots, I_n\}$ should be retrieved. This inherently multi-modal task can be defined as a scoring task $\Phi : \mathcal{I} \times \mathcal{T} \times \mathcal{D} \rightarrow \mathbb{R}$. While in standard CIR $\Phi$ is learned via supervised training, common zero-shot CIR (ZS-CIR) approaches sidestep this need by tuning specific modules to invert the query image into an associated text. Specifically, they learn an inversion function $\phi_i : \mathcal{I} \rightarrow \mathcal{Z}$ mapping a given query image to a pre-defined text-token embedding space $\mathcal{Z}$. Practically, $\phi_i$ is trained over intermediate image representation of a specific image encoder $\Psi_I$ (Saito et al., 2023; Baldrati et al., 2023), often part of a large-scale pre-trained vision-language representation system such as CLIP (Radford et al., 2021). Template filling around the text modifier over the corresponding inverted embedding $\text{inv}_q = \phi_i(\Psi_I(Q))$ is then used to aggregate the information into one target caption (e.g. "a photo of {$\text{inv}_q$} that {$t$}"). This target caption is then used for target image retrieval by VLMs like CLIP, encoding it using the associated pre-trained text encoder $\Psi_T$ that projects the target caption and candidate images $I \in \mathcal{D}$ into a shared, searchable embedding space. The respective matching score is then $\texttt{cos\_sim}(\Psi_I(I), \Psi_T(\text{inv}_q))$ with cosine similarity $\texttt{cos\_sim}$.

While promising, such a pipeline exhibits certain shortcomings, in that one i) needs to train a specific inversion module $\phi_i$ dependent on the chosen VLM and a separate image-caption dataset, ii) text embedding vectors can not be ensured to be human-understandable and cannot be verified as a correct description of the image, and iii) using rigid template filling does not allow for free-form textual representations and semantically flexible target captions.

### 3.2 COMPOSITIONAL IMAGE RETRIEVAL THROUGH VISION-BY-LANGUAGE

We can alleviate all the aforementioned shortcomings through Compositional Image Retrieval through Vision-by-Language (CIReVL) - a simple approach that recombines existing and publicly available pre-trained VLMs and LLMs. Similar to existing ZS-CIR methods (Saito et al., 2023; Baldrati et al., 2023), we build on CLIP as our retrieval system, however in a fashion that operates entirely independent on the particular CLIP model choice. We also assume access to pre-trained captioning models, s.a. readily available BLIP (Li et al., 2022; 2023) or CoCa (Yu et al., 2022), to provide a textual caption for a given image. Finally, we leverage a LLM for textual reasoning (Huang & Chang, 2023), available e.g. through Llama (Touvron et al., 2023), Vicuna (Chiang et al., 2023) or the GPT-framework (Brown et al., 2020). We visualize our framework in Fig. 1.

**From text embeddings to captions.** Textual inversion has two main issues. First, it relies on a specifically trained image-to-token-embedding mapping, tailored to image representations produced by a pre-defined, pre-trained encoder (e.g. CLIP). Second, predicted inversion tokens have no guarantee to be human-understandable. Both shortcomings are created by the learned token-generation module, and can thus be tackled by replacing it with an alternative system. As traditional token inversion methods rely on the existence of large-scale pre-trained models, a natural alternative is to find solutions from the larger corpus of large-scale pre-trained models. Specifically, by replacing trained textual inversion functions with an available image captioning system $\Psi_C$ through pre-trained generative VLMs such as BLIP, a simple solution for both problems can be found. Specifically, given a query $Q$, we obtain its textual representation as $c_q = \Psi_C(Q) \in \mathcal{T}$. As $c_q$ lives in the natural language text domain $\mathcal{T}$, it remains possible for humans to reason over - allowing the user clearer insights into the retrieval process and allowing for possible intervention, as we explore in §4.3.

**From templates to reasoning targets.** Using directly image captions $c_q$ is not sufficient as it does not incorporate the essential context provided by the text modifier $t$. While a simple pre-defined template may recombine these in specific, fixed ways, they have no flexibility to account for different forms of textual modifiers and caption-forms most suitable to a particular task at hand. To address this issue, we make use of the reasoning capabilities of existing LLMs. Rather than combining $c_q$ and $t$ in a fixed way using template filling, our goal is to obtain a unified target caption that models the hypothetical effect of $t$ on the query image $Q$ as a change in the thus resulting image caption $c_q$. This can be done in various ways, but we found simple prompts $p$ to encode sufficient problem context already (see §A for more details). In particular, given a LLM of choice $\Psi_R$, we generate an instruction-alterated image caption as $c_q^t = \Psi_R(p \circ c_q \circ t)$, which queries the LLM with a concatenation of the base prompt $p$, the generated image caption $c_q$ (prepended with `"Image Content:"`), and the instruction $t$ (prepended with `"Instruction:"`)[1]. Example queries and outputs are provided in §4.3. Note that the construction of the prompt is a one-time process requiring minimal effort and annotation, which we found to translate well across all problems. In addition, as the instruction-conditioning happens entirely in the language domain, a human-in-the-loop can fully reason about the impact of the instruction on the retrieval process by comparing $c_q$ and $c_q^t$ w.r.t. $t$.

**Compositional image retrieval.** Given the adapted caption $c_q^t$, `CIReVL` encodes the image-search database $\mathcal{D}$ alongside $c_q^t$ using a VLM (e.g. CLIP). The retrieved target $I_q^t$ is thus given as

$$I_q^t = \underset{I \in \mathcal{D}}{\text{argmax}} \frac{\Psi_I(I)^{\mathsf{T}} \Psi_T(c_q^t)}{||\Psi_I(I)|| \cdot ||\Psi_T(c_q^t)||}, \tag{1}$$

where the final selected target image is the one most similar to the generated target caption. As the image retrieval system is only introduced and utilized *after* the combination of query image and instruction, it is entirely detached and modular. Consequently, it can be easily exchanged with other ones depending on practical requirements and the desired trade-off between efficiency and efficacy. Overall, this results in a ZS-CIR pipeline in which compositions are human understandable as it operates entirely in the language domain, and the retrieval process exists as a clearly separated module, without requiring training of any mapping function on top of it.

## 4 EXPERIMENTS

We first provide the experimental details in §4.1, before showcasing the results of our `CIReVL` in four different ZS-CIR tasks in §4.2. Finally, we provide an in-depth analysis of our method in §4.3, highlighting it's capacity as well as the impact of the various components.

### 4.1 IMPLEMENTATION DETAILS

For our experiments we use PyTorch (Paszke et al., 2019), extending the public codebase of Baldrati et al. (2023), and using clusters of NVIDIA V100 and A100s. We experiment with different ViT-variants (Dosovitskiy et al., 2021) of CLIP, with weights taken from the official implementation in (Radford et al., 2021). We use the OpenCLIP (Ilharco et al.) models for the analysis on scaling laws. As captioner, we leverage the open-source BLIP-2 (Li et al., 2023) with a Flan-T5XXL

---

[1]For possible adaptation without training, in-context learning (see e.g. Dong et al. (2023)) can be leveraged.

Table 1: **Comparison on CIRCO and CIRR Test Data.** On CIRCO, `CIReVL` significantly outperforms even adaptive methods across retrieval models, while it achieves competitive results on CIRR despite the noise in the benchmark. Its modularity allows for simple further scalability for additional gains. (*) ViT-G/14 uses OpenCLIP weights (Ilharco et al.).

| CIRCO + CIRR → | | CIRCO | | | | CIRR | | | | | | |
|---|---|---|---|---|---|---|---|---|---|---|---|---|
| Metric | | mAP@k | | | | Recall@k | | | | $R_s$@k | | |
| Arch | Method | k=5 | k=10 | k=25 | k=50 | k=1 | k=5 | k=10 | k=50 | k=1 | k=2 | k=3 |
| ViT-B/32 | Image-only | 1.34 | 1.60 | 2.12 | 2.41 | 6.89 | 22.99 | 33.68 | 59.23 | 21.04 | 41.04 | 60.31 |
| | Text-only | 2.56 | 2.67 | 2.98 | 3.18 | 21.81 | 45.22 | 57.42 | 81.01 | 62.24 | 81.13 | 90.70 |
| | Image + Text | 2.65 | 3.25 | 4.14 | 4.54 | 11.71 | 35.06 | 48.94 | 77.49 | 32.77 | 56.89 | 74.96 |
| | PALAVRA | 4.61 | 5.32 | 6.33 | 6.80 | 16.62 | 43.49 | 58.51 | 83.95 | 41.61 | 65.30 | 80.94 |
| | SEARLE | 9.35 | 9.94 | 11.13 | 11.84 | 24.00 | 53.42 | 66.82 | 89.78 | 54.89 | 76.60 | 88.19 |
| | **CIReVL** | 14.94 | 15.42 | 17.00 | 17.82 | 23.94 | 52.51 | 66.0 | 86.95 | 60.17 | 80.05 | 90.19 |
| ViT-L/14 | Pic2Word | 8.72 | 9.51 | 10.64 | 11.29 | 23.90 | 51.70 | 65.30 | 87.80 | - | - | - |
| | SEARLE | 11.68 | 12.73 | 14.33 | 15.12 | 24.24 | 52.48 | 66.29 | 88.84 | 53.76 | 75.01 | 88.19 |
| | **CIReVL** | 18.57 | 19.01 | 20.89 | 21.80 | 24.55 | 52.31 | 64.92 | 86.34 | 59.54 | 79.88 | 89.69 |
| ViT-G/14* | **CIReVL** | 26.77 | 27.59 | 29.96 | 31.03 | 34.65 | 64.29 | 75.06 | 91.66 | 67.95 | 84.87 | 93.21 |

Table 2: **Comparison on FashionIQ Test Data.** `CIReVL` is able to significantly outperform adaptive methods across all Fashion-IQ sub-benchmarks, with its inherent modularity allowing for further simply scaling to achieve additional large gains. (*) OpenCLIP weights (Ilharco et al.).

| Fashion-IQ → | | Shirt | | Dress | | Toptee | | **Average** | |
|---|---|---|---|---|---|---|---|---|---|
| Backbone | Method | R@10 | R@50 | R@10 | R@50 | R@10 | R@50 | R@10 | R@50 |
| ViT-B/32 | Image-only | 6.92 | 14.23 | 4.46 | 12.19 | 6.32 | 13.77 | 5.90 | 13.37 |
| | Text-only | 19.87 | 34.99 | 15.42 | 35.05 | 20.81 | 40.49 | 18.70 | 36.84 |
| | Image + Text | 13.44 | 26.25 | 13.83 | 30.88 | 17.08 | 31.67 | 14.78 | 29.60 |
| | PALAVRA | 21.49 | 37.05 | 17.25 | 35.94 | 20.55 | 38.76 | 19.76 | 37.25 |
| | SEARLE | 24.44 | 41.61 | 18.54 | 39.51 | 25.70 | 46.46 | 22.89 | 42.53 |
| | **CIReVL** | 28.36 | 47.84 | 25.29 | 46.36 | 31.21 | 53.85 | 28.29 | 49.35 |
| ViT-L/14 | Pic2Word | 26.20 | 43.60 | 20.00 | 40.20 | 27.90 | 47.40 | 24.70 | 43.70 |
| | SEARLE | 26.89 | 45.58 | 20.48 | 43.13 | 29.32 | 49.97 | 25.56 | 46.23 |
| | **CIReVL** | 29.49 | 47.40 | 24.79 | 44.76 | 31.36 | 53.65 | 28.55 | 48.57 |
| ViT-G/14* | **CIReVL** | 33.71 | 51.42 | 27.07 | 49.53 | 35.80 | 56.14 | 32.19 | 52.36 |

language model (Chung et al., 2022). Ablations consider also BLIP (Li et al., 2022) and CoCa (Yu et al., 2022). As LLM we use gpt-3.5-turbo Brown et al. (2020), but we experiment also with Vicuna13B (Chiang et al., 2023), Llama2-70B (Touvron et al., 2023), and GPT-4 (OpenAI, 2023).

**Datasets and Baselines.** We use the CIRR (Liu et al., 2021), CIRCO (Baldrati et al., 2023), FashionIQ-(Wu et al., 2021) and GeneCIS (Vaze et al., 2023) datasets which have all been used for CIR. CIRR, the first natural image dataset for CIR, suffers from false negatives (Baldrati et al., 2023), since it has only a single target image annotated. The recently introduced CIRCO dataset ameliorates this by having multiple positive images for each query. The GeneCIS dataset (sourced from MS-COCO (Lin et al., 2014) and Visual Attributes in the Wild (Pham et al., 2021)) introduces four task variations, retrieving or changing a specific attribute or object. Due to space constraints, we report results on the ImageNet Domain benchmark (Saito et al., 2023) in the appendix. Fashion-IQ is a benchmark that is focused around retrieval in fashion settings. Following the original benchmarks, we use Recall@k as the metric on the CIRR, GeneCIS, and Fashion-IQ. On the CIRCO dataset, since there are multiple positives, we use the mean average precision (mAP@k). We use the 'image-only', 'text-only' and 'image+text' to denote directly performing retrieval with CLIP using only the reference image, modifying instruction, as well as averaging the embeddings for the reference image and modifying text. PALAVRA (Cohen et al., 2022), Pic2Word (Saito et al., 2023), SEARLE (Baldrati et al., 2023) are textual inversion methods either designed or adapted for ZS-CIR.

## 4.2 ZS-CIR BENCHMARK COMPARISONS

**CIRCO.** Our results in Tab. 1 list the performance on the hidden test set of CIRCO, accessible through the submission server provided Baldrati et al. (2023). As can be seen, using the default ViT-B/32 and ViT-L/14 CLIP variants, our approach significantly outperforms methods like Pic2Word, SEARLE and PALAVRA (Cohen et al., 2022). For instance, on ViT-L/14, we achieve a mAP@5 of 18.57% - notably improving over the 11.68% by the best performing, *trained* alternative SEARLE,

Table 3: **Comparison on GeneCIS Test Data.** `CIReVL` is able to significantly outperform adaptive methods across all Fashion-IQ sub-benchmarks, with its inherent modularity allowing for further simply scaling to achieve additional large gains. (*) OpenCLIP weights (Ilharco et al.).

| GeneCIS → | | Focus Attribute | | | Change Attribute | | | Focus Object | | | Change Object | | | Average |
|---|---|---|---|---|---|---|---|---|---|---|---|---|---|---|
| Backbone | Method | R@1 | R@2 | R@3 | R@1 | R@2 | R@3 | R@1 | R@2 | R@3 | R@1 | R@2 | R@3 | R@1 |
| RN50x4 | Image Only | 17.7 | 30.9 | 41.9 | 11.9 | 20.8 | 28.8 | 9.3 | 18.2 | 26.2 | 7.2 | 16.7 | 24.9 | 11.5 |
| | Text Only | 10.2 | 20.5 | 29.6 | 9.5 | 17.6 | 26.4 | 6.5 | 16.8 | 22.4 | 6.2 | 13.9 | 21.4 | 8.1 |
| | Image + Text | 15.6 | 26.3 | 37.1 | 12.6 | 22.9 | 32.0 | 10.8 | 21.0 | 31.2 | 11.3 | 21.5 | 30.3 | 12.6 |
| | Combiner (CIRR) | 15.1 | 27.7 | 39.8 | 12.1 | 22.8 | 31.8 | 13.5 | 25.4 | 36.7 | 15.4 | 28.0 | 39.6 | 14.0 |
| | Combiner (CC3M) | 19.0 | 31.0 | 41.5 | 16.6 | 27.5 | 36.5 | 14.7 | 25.9 | 36.1 | 16.8 | 29.1 | 39.7 | 16.8 |
| | **CIReVL** | 19.1 | 31.3 | 41.5 | 14.7 | 26.3 | 37.0 | 13.2 | 23.4 | 32.7 | 16.3 | 28.7 | 37.9 | 15.8 |
| ViT-B/32 | SEARLE | 18.9 | 30.6 | 41.2 | 13.0 | 23.8 | 33.7 | 12.2 | 23.0 | 33.3 | 13.6 | 23.8 | 33.3 | 14.4 |
| | **CIReVL** | 17.9 | 29.4 | 40.4 | 14.8 | 25.8 | 35.8 | 14.6 | 24.3 | 33.3 | 16.1 | 27.8 | 37.6 | 15.9 |
| ViT-L/14 | SEARLE | 17.1 | 29.6 | 40.7 | 16.3 | 25.2 | 34.2 | 12.0 | 22.2 | 30.9 | 12.0 | 24.1 | 33.9 | 14.4 |
| | **CIReVL** | 19.5 | 31.8 | 42.0 | 14.4 | 26.0 | 35.2 | 12.3 | 21.8 | 30.5 | 17.2 | 28.9 | 37.6 | 15.9 |
| ViT-G/14* | **CIReVL** | 20.5 | 34.0 | 44.5 | 16.1 | 28.6 | 39.4 | 14.7 | 25.2 | 33.0 | 18.1 | 31.2 | 41.0 | **17.4** |

Figure 2: Examples from the CIRCO validation set where our method retrieves the desired image. We see that our method is able to perform this task for a wide variety of modifier texts.

and more than doubling the performance of Pic2Word, which achieves $8.72\%$. These are strongly indicative results, as CIRCO constitutes the dataset with the cleanest annotations and - unlike other datasets in this field - the inclusion of multiple positives for an inherently ambiguous problem, as textual modifications of an image do not only have a single solution. These strong results thus provide key evidence of the efficacy of our training-free, vision-by-language approach for ZS-CIR.

**CIRR.** For the hidden CIRR test set (accessible using a test server, see e.g. Liu et al. (2021)), we provide results in Tab. 1. As previously noted, this dataset is very noisy, with results primarily dependent on the modifying instruction, and the actual reference image having much less relation to the target image (Saito et al., 2023; Baldrati et al., 2023). Still, even on this benchmark, we are able to match the performance of prior ZS-CIR methods (e.g. $R@1 = 24.55\%$ for our method versus $24.24\%$ for SEARLE), without requiring any form of problem-specific training. The CIRR benchmark also provides another evaluation where the correct image has to be retrieved from 6 curated samples. In this evaluation, our results surpass prior work by a significant margin ($R_s@1 = 60.17\%$ versus $54.89\%$ for SEARLE), underscoring the versatility of our method.

**Fashion-IQ.** We provide the results on the validation set of the Fashion-IQ benchmark in Tab. 2. We see that `CIReVL` is able to outperform prior zero-shot methods by significant margins (average $R@10$ of $28.55\%$ versus $25.56\%$ for SEARLE). Additionally, we also see the benefits of scaling the model size, with the average $R@10$ improving to $32.19\%$. This also underscores how our approach is able to adapt to diverse domains with minimal changes.

**GeneCIS.** The versatility of `CIReVL` is further underlined when transferring it to the GeneCIS benchmark. Unlike CIRCO and CIRR, modifiers consists of a single word that has a different interpretation for each task, e.g. focusing/changing a particular attribute or object. In this case, without re-training any method, simple prompt modifications allow for convincing performance. In particular, for the "focus"-tasks, we simply task the LLM to *retain the attribute/object listed in the instruction*. For "change"-tasks, we simply ask it to *replace the corresponding object* in the caption. As shown in Tab. 3, `CIReVL` nearly matches the performance of Combiner trained on a large filtered

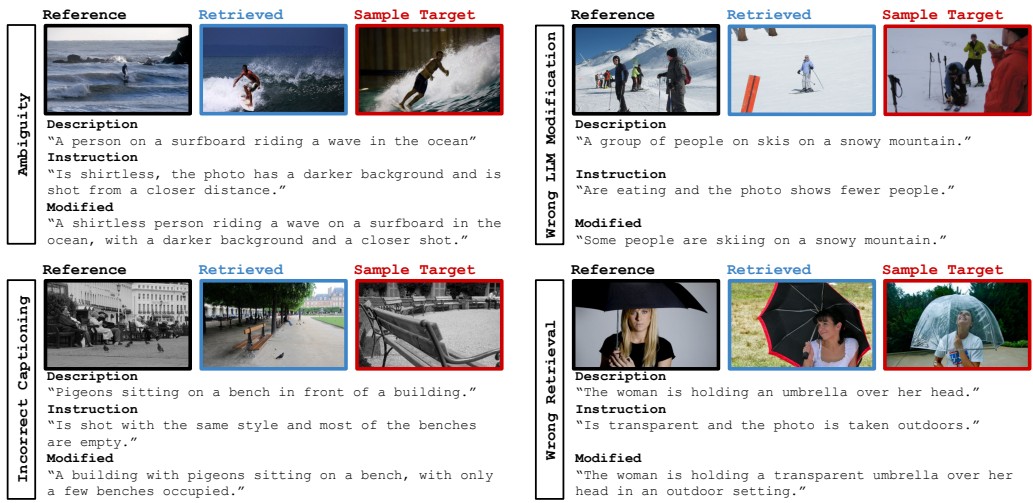

Figure 3: We analyze failure cases of our approach. Due to the interpretable nature, we can easily attribute errors to captioning, reasoning or the text-image retrieval. We also find examples where the model is penalized despite a plausible retrieval due to insufficient annotations in the dataset.

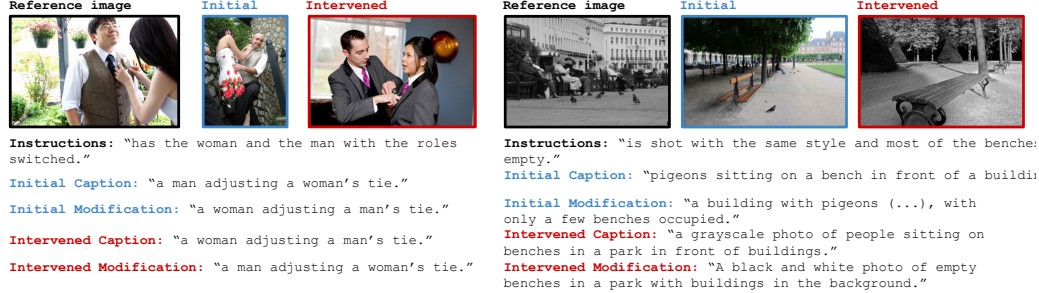

Figure 4: We demonstrate the possibility of user interventions to enhance the performance of our method. For instance, by fixing the mistakes in the generated caption, we are able to correctly retrieve the desired image without having to make any other changes.

version of the CC3M dataset (Sharma et al., 2018) (16.8% Average $R@1$ versus 15.9%), and surpass the Combiner model finetuned on the CIRR dataset as well as other baselines. This is notable given the diversity, but also specificity of the tasks, ranging from settings resembling standard image retrieval (e.g. "focus attribute") to traditional CIR tasks instead (e.g. "change object").

## 4.3 ABLATION STUDY AND PERFORMANCE ANALYSIS

In this section, we conduct a large number of analyses to provide a better understanding of our proposed method through ablations and qualitative examples, and provide insights into performance bottlenecks. This also includes scalability studies, and insights into the benefits of human understandability via natural language operations, further enabling the possibility to perform interventions on the retrieval process in order to tackle failure cases.

**On the importance of textual reasoning.** Since our proposed method heavily relies on the LLM to convert textual inputs to target captions, we study several LLMs to understand the effect they play on the final performance. The results on the CIRCO validation set in Table 4 illustrate that the reasoning is critical to the overall performance. Most notably, we perform a sanity-check where the generated caption is used to fill a template as done in prior works (Saito et al., 2023; Baldrati et al., 2023). This method ('Captioning') aims to test the necessity of using a LLM for textual reasoning on this task. We see that this method is outperformed by all the LLMs that we tested (mAP@5 of 9.22%). This highlights the necessity of performing textual reasoning to generate the final target caption, and the limits of static templates. For instance, both GPT-3.5-turbo and GPT-4 achieve strong results (mAP@5 of 13.33% and 15.63% respectively), with the improved capabilities of GPT-

Table 4: **Analysis of the impact of LLM and captioner choice.** Comparison between different LLMs on CIRCO validation reveals a positive relation between performance and reasoning capacities. For captioning models, most recent off-the-shelf models achieve strong results.

| Arch | Captioner | LLM | mAP@5 | mAP@10 | mAP@25 | mAP@50 |
|------|-----------|-----|-------|--------|--------|--------|
| ViT-B/32 | BLIP-2 | - | 9.22 | 10.05 | 11.44 | 12.08 |
| | | LLama2-70B | 10.22 | 10.60 | 11.80 | 12.39 |
| | | Vicuna-13B | 12.65 | 13.17 | 14.48 | 15.20 |
| | | GPT-3.5-Turbo | 13.33 | 14.16 | 15.74 | 16.35 |
| | | GPT-4 | 15.63 | 16.31 | 17.02 | 18.12 |
| ViT-B/32 | BLIP | GPT-3.5-Turbo | 13.1 | 13.33 | 15.08 | 15.92 |
| | BLIP-2 | | 13.33 | 14.16 | 15.74 | 16.35 |
| | CoCa | | 13.64 | 13.37 | 15.10 | 15.90 |

4 offering significant performance gains on top. When moving to publicly accessible LLMs such as Llama2-70B and Vicuna-13B (10.22% and 12.65%), we find a fair drop in performance. For cases where access to closed-source APIs such as GPT-3.5-turbo or GPT-4 is restricted, alternative usage of public LLMs can thus still offer significant benefits, particularly when evaluating the differences between Vicuna-13B and GPT-3.5-turbo.

**On the importance of captioning quality.** As our CIR framework builds on the generated reference image caption, we also ablate the importance of the correct captioning model choice in Tab. 4, which reports results on the CIRCO validation split for faster turnaround. Interestingly, we find that most state-of-the-art public captioning models perform similarly well, with minor differences between BLIP versions, and slightly improved performance of CoCa. To retain the generality of our approach, we stick with BLIP-2 with a Flan-T5 language model since the usage of a LLM decoder makes it a more generic tool usable across a variety of domains.

**Reasoning Sanity Check by Measuring Text-Image Alignment.** We conduct a quantitative evaluation of our reasoning component by assessing the text-image alignment in the LLM-modified descriptions using TIFA (Hu et al., 2023). Unlike simpler metrics such as CLIP score, TIFA offers a more accurate measure of cross-modal alignment by converting it into a series of question-answering tasks. Results are presented in Fig. 5. Comparing the alignment between our generated modified descriptions and the generated plain image captions, we

Figure 5: We use TIFA (Hu et al., 2023) to compare alignment between LLM descr. & base caption and target image, and LLM descr. to target- and CLIP-retrieved image. It shows the impact of LLM reasoning over base captions, and CLIP retrieval as essential bottleneck.

clearly see much higher and more consistent average alignment through our modification. In addition, when looking at the alignment of our modified caption between the ground truth image and the actually retrieved image by CLIP (ViT-B/32) in Fig. 5, we see that the alignment of the retrieved image is actually notably lower than that of the ground truth image. Thus, while the CLIP similarity is higher for the retrieved image, its actual alignment with the modified caption is lower. This means that the standard CLIP backbone, even if the modified caption actually aligns well with the ground truth image, can often fail to retrieve it. This indicates that the CLIP retrieval is a severe bottleneck (Thrush et al., 2022; Hsieh et al., 2023; Kamath et al., 2023), as the captioning and reasoning together generate valid target captions that a sub-optimal retrieval system cannot match.

**Investigating Scaling Laws for ZS-CIR.** A critical benefit of the modular nature of our method is that we can replace and scale each component, without re-training. In particular, we can simply utilize our base model used for our benchmark comparisons in §4.2, and utilize a different CLIP model for the final retrieval process. This allows us to investigate scaling laws as studied and investigated in e.g. Kaplan et al. (2020); Caballero et al. (2023); Cherti et al. (2023) for the particular problem of CIR, and investigate if scale can address the retrieval bottleneck. For this, we leverage CLIP variants provided through the OpenCLIP project (Ilharco et al.), with models ranging from around 150M to-

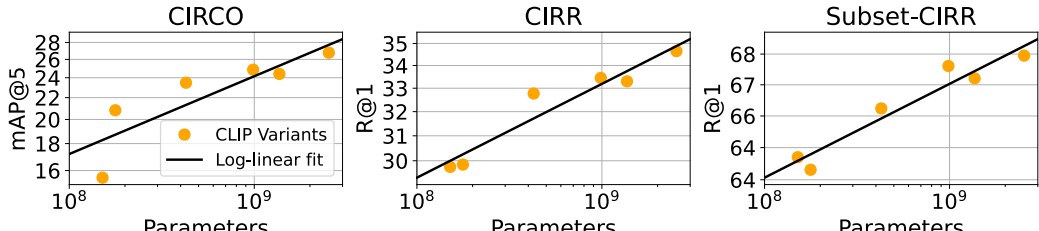

Figure 6: Leveraging our modular `CIReVL` to study scaling laws. For model availability and consistency, we use LAION-2B pretrained CLIP models provided through OpenCLIP (Ilharco et al.).

tal parameters to around 2.5B. On both CIRCO and CIRR, we see a clear log-linear relationship between the model capacity and the performance as a result of simply upgrading our retrieval model in size. This clearly highlights the impact of scale even to the complex setting of ZS-CIR, and allows us to partly break the retrieval bottleneck shown above. As we can easily plug-and-play different retrieval models, it also allows us to scale our overall pipeline, allowing us to boost the overall CIR scores on the CIRR dataset up to $R@1 = 34.64\%$ (c.f. $24.55\%$ for the highest reported SEARLE score), and up to $mAP@5 = 26.77\%$ on CIRCO (c.f. $11.68\%$ for the highest reported SEARLE score). These results provide clear evidence of our method allowing for a simple transition between efficiency (with a smaller retrieval model) and efficacy with a larger one when needed. We also find that our insights partly contrast those in Vaze et al. (2023), where only minimal gains from scaling can be observed. Our results show that by incorporating conditioning through textual reasoning, `CIReVL` better leverages the benefits of large retrieval models without expensive re-training.

**Qualitative Examples.** Fig. 2 visualizes successful CI-retrievals with instructions impacting different semantic elements of the reference image such as viewpoint, color, object counts, background changes, object insertion, object adaptation or picture manipulations such as zoom. This provides further indication about the diverse applicability of our setup. Following that, Fig. 3 visualizes exemplary failure cases. As our CIR process operates primarily in the language domain, it becomes easy to understand the particular failure cases. For instance, we see some cases where the generated *Caption* either does not correctly describe the image, or does not focus on the relevant aspect in the image (e.g. ignoring the grayscale aspect of the image and focusing on the pigeons). We also observe cases where the LLM does not generate an accurate description of the target image. Additionally, there are also cases where despite the captioning and reasoning working correctly, the CLIP model does not *retrieve* an accurate image. Finally, we also see ambiguous failure cases where the retrieved target is debatable correct, but not labeled as such due to incomplete annotations (*Ambiguity*).

**Performing User Interventions.** Fortunately, by being able to break down the retrieval process, we can easily perform user interventions when needed. Simple example scenarios are shown in Fig. 4, where the default modified caption results in incorrect retrievals. As a user, intervention can happen at different stages of the retrieval process, modifying e.g. generated captions, utilized instructions or the final LLM-generated caption. As a proof of concept, we showcase how intervention on the base caption level can already result in correct search queries on the image database. Simple human intervention can easily determine if a generated caption is incorrect, and simply replace it with an alternative. The rest of the pipeline can simply continue operating on top of this intervention. As can be seen in Fig. 4, intervening on the caption-level can already rectify various errors. This once again sharply contrasts with existing work that relies on an end-to-end pipeline with a rigid query template, where there is no possibility to alter the retrieval process.

## 5 CONCLUSION

In this work, we present a novel, training-free approach for Zero-Shot Compositional Image Retrieval (CIR). Utilizing off-the-shelf pre-trained models, our method not only achieves strong performance across multiple CIR benchmarks but also, in some cases, doubles the performance of existing state-of-the-art methods. The method's inherent interpretability allows for user intervention, adding an extra layer of flexibility. We also explore the impact of scaling laws on our method, revealing that scaling the text-image retrieval component can substantially boost task performance. Collectively, these contributions set the stage for future research in training-free Compositional Image Retrieval.

ACKNOWLEDGEMENTS

This work was supported by DFG project number 276693517, by BMBF FKZ: 01IS18039A, by the ERC (853489 - DEXIM), by EXC number 2064/1 – project number 390727645, and by the MUR PNRR project FAIR - Future AI Research (PE00000013) funded by the NextGenerationEU. Shyamgopal Karthik and Karsten Roth thank the International Max Planck Research School for Intelligent Systems (IMPRS-IS) for support. Karsten Roth would also like to thank the European Laboratory for Learning and Intelligent Systems (ELLIS) PhD program for support. The authors also thank Vishaal Udandarao for valubale feedback and Thomas Hummel for help with visuals.

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

## A   ADDITIONAL IMPLEMENTATION DETAILS

The prompt that we use draws upon the one used by Liu et al. (2023) for curating a dataset curation. Here, we write a similar prompt for the LLM to generate the edited description. We list the full prompt below:

```
"I have an image.  Given an instruction to edit the image,
carefully generate a description of the edited image.  I
will put my image content beginning with 'Image Content:'.
The instruction I provide will begin with 'Instruction:'.
The edited description you generate should begin with
'Edited Description:'.  Each time generate one instruction
and one edited description only."
```

Table 5: Comparison with supervised baselines on CIRR and FashionIQ validation sets. Combiner-FIQ and Combiner-CIRR denote the models from Baldrati et al. (2022) trained on FashionIQ and CIRR, respectively. We see that zero-shot methods generalize better across multiple datasets.

| Method | CIRR | | | | FashionIQ | |
| --- | --- | --- | --- | --- | --- | --- |
| | R@1 | R@5 | R@10 | R@50 | R@10 | R@50 |
| Combiner-FIQ Baldrati et al. (2022) | 19.88 | 48.05 | 61.11 | 85.51 | **32.96** | **54.55** |
| Combiner-CIRR Baldrati et al. (2022) | **32.24** | **65.46** | **78.21** | **95.19** | 20.91 | 40.40 |
| **CIReVL** | 29.76 | 59.69 | 71.39 | 90.34 | 28.29 | 49.35 |

## B   COMPARISON WITH SUPERVISED BASELINES

We provide a comparison to supervised CIR methods in Tab. 5. In particular, we compare our results against the Combiner architecture (Baldrati et al., 2022). We train the Combiner model on both the Fashion-IQ and CIRR datasets. We see that while the Combiner model achieves strong results when it has been fine-tuned on a particular dataset, it performs poorly on the other benchmarks. In contrast, zero-shot methods are able to achieve strong performance across benchmarks, highlighting the generalization capabilities of these models.

## C   EXPERIMENTS ON IMAGENET DOMAIN CONVERSION

We also test CIReVL on the ImageNet domain conversion experiment proposed in (Saito et al., 2023). Here we use images from 200 classes of the original ImageNet dataset (Russakovsky et al., 2015) as query, and for retrieval images of the same object but in the specified domain from ImageNet-R (Hendrycks et al., 2021) Unlike the previous benchmarks, the task is to simply retrieve an image of the appropriate domain for the same semantic object category (i.e a cartoon of a goldfish, with a natural goldfish reference image and the modifier `"cartoon"`). This requires no reasoning over image semantics, as the modifier affects an independent domain change, with significant improvements over Pic2Word or Combiner already be achieved by leveraging the final description `"a domain of a caption"`, as seen in Tab. 6. Our model in parts more than double the performance of Pic2Word on this task (e.g. $R@1 = 19.2\%$ versus $8.0\%$ for a conversion to the cartoon domain). These findings mainly reiterate that combining off-the-shelf pre-trained models - here CLIP and BLIP-2 - can be more effective than customized models operating on top of pre-trained models.

Table 6: **Evaluation on ImageNet Domain Conversion experiment proposed in Saito et al. (2023).** The goal is to retrieve an appropriate domain of the object specified in the query image.

| | Cartoon | | Toy | | Origami | | Sculpture | |
|---|---|---|---|---|---|---|---|---|
| | R@10 | R@50 | R@10 | R@50 | R@10 | R@50 | R@10 | R@50 |
| Image-only | 0.3 | 4.5 | 0.2 | 1.8 | 0.6 | 5.7 | 0.3 | 4.0 |
| Text-only | 0.2 | 1.1 | 0.8 | 3.7 | 0.8 | 2.4 | 0.4 | 2.0 |
| Image+Text | 2.2 | 13.3 | 2.0 | 10.3 | 1.2 | 9.7 | 1.6 | 11.6 |
| Combiner (CIRR) (Baldrati et al., 2022) | 6.1 | 14.8 | 10.5 | 21.3 | 7.0 | 17.7 | 8.5 | 20.4 |
| Pic2Word (Saito et al., 2023) | 8.0 | 21.9 | 13.5 | 25.6 | 8.7 | 21.6 | 10.0 | 23.8 |
| `CIReVL` | **19.2** | **42.8** | **30.2** | **41.3** | **22.2** | **43.1** | **23.4** | **45.0** |

