# OpenReview forum: "Vision-by-Language for Training-Free Compositional Image Retrieval"
_ICLR.cc/2024/Conference — ICLR 2024 poster_

### Official Review · Reviewer_Tg9t · 2023-10-31

**Soundness:** 3 good
**Presentation:** 4 excellent
**Contribution:** 2 fair
**Rating:** 5
**Confidence:** 5

**Summary:**

This paper proposes a training-free framework for zero-shot image retrieval, by leveraging the pretrained vision-language model and large language model. VLM is used to obtain image caption for query image, and the caption together with the text modifier is fed into LLM to get the target caption; then the target caption is used to do cross-modal retrieval with the help of VLM in database images. This framework also allows scalability and human intervention to improve the retrieval performance due to its modularity and human-understandability.

**Strengths:**

1. The proposed framework is training-free and achieves promising retrieval results.
2. The retrieval accuracy could be improved by adopting better off-the-shelf VLM and LLM, due to the modularity of the framework.
3. Since the query is mainly processed in language domain, the framework is human-interpretable and could also improve the retrieval accuracy by involving human intervention.

**Weaknesses:**

1.	The method itself is very intuitive and not quite novel, which is more like an engineering extension of existing models like VLM and LLM. Additionally, the proposed method does not achieve better results on all evaluation datasets. In Table 1, CIReVL underperforms SEARLE on CIRR dataset in terms of CIRR, which shows the unstability.
2.	This work only adopts datasets of everyday life and natural scenes (CIRR, CIRCO, GeneCIS) as evaluation. It is necessary to include datasets of various domains, such as fashion domain datasets like fashioniq and fashion200k to evaluate the generalization ability. Furthermore, for GeneCIS dataset, the comparison against other ZSCIR methods such as Pic2word and SEARLE should be included for complete comparison.
3.	The proposed framework is not very time-efficient since for each query, there will be two auto-regressive processes: query image captioning and target caption generation, which are very time-consuming. Both processes involve large models like VLM and LLM, which may further lead to low efficiency for image search. Therefore, it is necessary to compare the retrieval efficiency (time) with the previous methods.

**Questions:**

The same as weaknesses.

---

> ### Author Response · Authors · 2023-11-14
> **Response to Review**
>
> We thank the reviewer for their thoughtful comments. We appreciate that they find our results promising, and our method interpretable. Below, we address the concerns raised:
>
> ---
>
> __1. Novelty and results on CIRR:__
> Indeed, our method is an application of off-the-shelf VLMs and LLMs. However, their particular recombination and reuse to transfer it to the difficult task of compositional image retrieval is novel. This is particularly interesting, as unlike previous approaches that leverage off-the-shelf VLMs as well and have to train on large amounts of data, our approach requires no training at all, while the modular structure and operation in the language domain makes our approach interpretable and intervenable as well.
>
> This strongly contrasts our approach from existing ones! In addition, as our scaling laws study shows, our setup can be easily scaled up to account for modular improvements over time, without having to retrain. As such, we strongly believe that our work is an important contribution to the field of compositional image retrieval.
> In addition, we outperform existing, trained methods on CIRCO and Fashion-IQ significantly, while on CIRR (with noisy supervision, lack of multiple positive annotation, etc. as shown in previous works and noted in the paper), we are able to match previously obtained results
>
> ---
>
> __2. Results on Fashion benchmarks:__
>
> We thank the reviewer for the suggestion. We have now updated our paper with the results on the Fashion-IQ benchmark. For the detailed results, we refer to our shared comment. In summary, we find that CIReVL significantly outperforms existing methods on all three subtasks, e.g. improving average Recall@10 by nearly 6pp (28.29% versus 22.89% for SEARLE) with a ViT B/32 backbone. In addition, the simple scalability allows us to hot-swap the retrieval model to achieve additional 4pp (increasing from 28.29% to 32.19%).
>
> ---
>
> __3. Inference time:__
> Indeed, while existing methods also utilize large VLMs, the use of distinct captioning and reasoning modules elicits additional runtime. On our hardware setup (V100 / A100 GPUs) when compared to trained methods, we find inference times to increase by ~1s. However, the modularity also allows our setup to directly benefit from efficiency changes within each module (as e.g. seen in the cost & inference time reductions for LLMs over the past year), allowing one system to remain operational for a long period of time. This stands in particular contrasts to other ZS-CIR methods that require costly training to incorporate any pipeline changes or shifts in expected datasets (e.g. Pic2Word trains the mapping network on 8 GPUs on the complete CC3M dataset).

---

> > ### Comment · Reviewer_Tg9t · 2023-11-22
> >
> > Thanks for the authors' response. However, the novelty of this paper is not significant enough. Besides, the performance and inference time of the proposed method still limit its real application. Therefore, I will keep my rating.

---

> ### Author Response · Authors · 2023-11-22
>
> We thank the reviewer for their response, and would like to address two mentioned points:
>
> ---
>
> > (...) the novelty of this paper is not significant enough"
>
> It would be great if the reviewer could provide additional context for this statement, as in our reply, we do highlight that our method and framework is novel. Even if the single components themselves are not novel on their own, their particular recombination and reuse is. Other approaches also leverage large-scale, off-the-shelf pretrained models and existing network architectures.
> Unlike these approaches however, our setup does not require large amounts of additional training data on top.
>
> In addition to that, our method is first and foremost motivated by important practical constraint listed above, which significantly differentiates it from existing approaches, namely it being
> * interpretable,
> * intervenable,
> * easily extendable,
> * and providing better performance.
>
> In addition to that, it also provides an important sanity check for future method that do incorporate additional training data.
>
> ---
>
> > (...) the performance and inference time of the proposed method still limits its real application
>
> Large-scale models find ubiquitous usage in real world applications, ranging from LLM services to recent large-scale multimodal models. In addition, our performance significantly outperforms existing methods on all benchmarks but one (and comparable scores in that case), while not requiring any training. Furthermore, the ability to freely interchange models is a property that especially allows for more realistic deployment based on cost-performance trade-offs.

---

### Official Review · Reviewer_JtMi · 2023-10-31

**Soundness:** 3 good
**Presentation:** 2 fair
**Contribution:** 2 fair
**Rating:** 5
**Confidence:** 4

**Summary:**

This paper introduces CIReVL, a training-free pipeline for zero-shot compositional image retrieval by combining existing off-the-shelf foundation models, e.g., BLIP-2 for image captioning, GPT for text editing and CLIP for image retrieval. In addition, the specific and explicit description with captions, instead of text embeddings, facilitates human understanding over the retrieval process. Extensive experiments are conducted to validate the proposed method.

**Strengths:**

1.	The paper effectively integrates the VLM and LLM for ZS-CIR, offering a flexible and intervenable CIR system.
2.	The authors have conducted thorough experiments to validate the effectiveness of the proposed method, making the evaluation rather reliable.
3.	The figures in this paper are clear, effectively conveying the processing pipeline.

**Weaknesses:**

1.	Limited contribution: The proposed method appears to be a combination of several foundation models and simply employs the basic modelling capacity of these models (e.g., BLIP-2 for image captioning, GPT for text editing and CLIP for image retrieval), presenting a naïve and straightforward solution for CIR task. As a result, it is challenging to identify insightful and significant contributions to this field. The authors should conduct further in-depth research to enhance their contributions.
2.	Incorrect experimental results: The Recall@10 of CIReVL (ViT-G/14∗) on CIRR dataset in Table 1 is obviously incorrect.
3.	Inconsistent formatting of table data: In some cases, the data is presented with two decimal places, while in others, only one decimal place is used. Furthermore, there is a case with a mix of formatting in Table 3. The authors may revise the paper more carefully.
4.	The paper effectively combines the existing powerful VLM and LLM models. However, it would be better to provide a more insightful analysis of the caption and reason processes. For example, from Table 3, compared to utilizing different LLM models, the use of various state-of-the-art (SOTA) captioning models has a relatively minor impact on retrieval. Are there any potential explanations for this phenomenon? Does this mean the choice of caption model is not strict, as long as the model can catch the main object or attribute of images? I think it would be better to provide more analysis.
5.	The description about other works in Section 3.1 is a bit obscure for me who is not so familiar with the task but proficient other related tasks.
6.	Missing references, such as in ‘Similar to existing ZS-CIR methods’ in the first paragraph of Section 3.2.
7.	Experiments: (a) In Table 1, the author only presents the experimental results of ‘image only’ and ‘image+text’ methods for reference while missing that of the ‘text only’ method under the ViT-B/32 setting. However, the authors mention that the results on CIRR benchmark are primarily dependent on the modifying instruction while the actual reference image has much less relation to the target image, which indicates that the ‘text-only’ method can be an important reference for measuring the performance of the proposed method. Better to present the result of ‘text-only’ method and make a fair comparison and discussion. (b) In Table 1, the authors miss the results of ‘image only’, ‘text only’ and ‘image+text’ method for ViT-L/14 and ViT-G/14 settings. Better to include them for reference. (c). When evaluating on the GeneCIS benchmark, the authors do not specify the architecture of the vision backbone adopted in the experiment. Better to specify it clearly for reference.
8.	Discussions and evaluation on the potential limitations: The paper shows that the proposed method has several merits including free of training, good flexibility and scalability. However, it may also have some potential limitations. For example, (a) Inference costs and efficiency: The proposed method utilizes large VLMs and LLMs to conduct the image captioning, language reasoning and cross-modal retrieval during inference. Will it take more computational costs and have longer inference time compared with the previous methods? (b) Limitations of each module: Since the proposed method is composed of three different modules, the effect of each module plays an important role on the final retrieval results. For example, if the image caption module generates partial or false descriptions for the given images, the reasoning and retrieval process will be misled. Thus, it is necessary to analyze the potential negative impact brought by each module in an in-depth manner and quantify them if possible (which factor contributes more to the failure cases). (c) Compatibility of different modules: Since the proposed method conducts cross-modal retrieval by cascading three separate modules, the compatibility of different modules seems to be an important factor. For example, if the captions generated by LLMs have different styles with pretraining data of the VLMs, the VLMs used for cross-modal retrieval may produce some bias, which may hinder the final performance. Overall, it will be appreciated if the authors can present more in-depth discussions and evaluation on the potential limitations to fully demonstrate the properties of the proposed method.
9.	Writing: Some parts of the paper don’t flow well. The overall writing requires further improvement.

**Questions:**

1.	This paper seems to be a technical report with the application of current state-of-the-art foundation models. Although directly using existing models is convenient, it inevitably introduces errors in each processing step. How to address the cumulative errors resulting from these multiple models?
2.	Could the authors provide experimental results of CIR works with a supervised training pattern? Based on the current results in this paper, the overall zero-shot retrieval performance appears to be poor. Do those supervised methods also yield unsatisfactory performance? If so, it seems that more significant efforts are needed to address the fundamental issues in CIR task; If not, does the zero-shot approach still have its value?
3.	How you get the 6 curated samples for the Rs @K metric in Table 1?
4.	For reasoning, the experiments show LLM models obviously affect the performance. I wonder whether the same is true for the prompt template. Except for the prompt mentioned in Appendix A, were there any alternative prompts explored during the experimentation?

---

> ### Author Response · Authors · 2023-11-14
> **Response to Review**
>
> We thank the reviewer for their detailed comments, and the appreciation of our flexible and interpretable CIR system, thorough experimentation and clear visualization. We address their issues and feedback below.
>
> ---
>
> __1. Limited Contribution:__
> We agree with the reviewer that our approach is a combination of existing VLMs and LLMs. However, their particular recombination to transfer it to compositional image retrieval, in which previous works have always leveraged (large-scale) additional training data to make the system work, are to the best of our knowledge, novel insights. In addition to that, the recombination and particularly chosen modular structure of our proposal offers an approach to CIR which is i)  inherently interpretable unlike existing methods (which also leverage existing pretrained VLMs & LLMs), ii) can be easily scaled up and iii) allows for human intervention into the retrieval process - all properties, which are not existent in previous works which require additional training data! As such, we do believe that our paper provides a novel, practical contribution on compositional image retrieval.
>
> ---
>
> __2. & 3. Typo in the result and errors in table formatting:__
> We thank the reviewer for pointing out these errors, and have rectified these in the updated draft.
>
> ---
>
> __4. On the differing impact of captioning and LLM modules:__
> The reviewer rightly points out that particular choices in the image captioning module have less impact on the overall compositional image retrieval process as opposed to changes in the reasoning module (the choice of LLM). We do believe this to be primarily due to two reasons. First, the performance gaps between the tested captioning models is much less evident compared to the gap between available foundational LLMs (Llama versus GPT-4 have vastly different reasoning capabilities). This transitions to the second aspect, in that - as correctly noted - for all benchmarks, only a subset of crucial image properties have to be captured. This can be sufficiently done by all captioning models. Indeed, when looking at failure cases such as in Fig. 3, we find that a lack of captioning quality to be one of the main error sources - something which a notably better captioning model could fix in the future.
>
> Given the modularity of our setting, something that can be simple hot-swapped, as opposed to having to retrain a full pipeline from scratch!
> Finally, we believe that a more detailed investigation should involve the development of benchmarks which cannot be easily described through single, in parts simplistic captions. This is something we leave for future work to tackle.
>
> ---
>
> __5 & 6 & 7. Writing Suggestions, missing references in 3.2 and missing baseline references:__
> We thank the reviewer for their valuable feedback. We have updated our paper accordingly, by adding a more detailed breakdown of the compositional image retrieval task, as well as referencing related ZS-CIR methods in the beginning of Section 3.2 (particularly Pic2Word & SEARLE). In addition, we have included the “text-only”, “image-only” and “image+text” baseline across our experimental results. We do note that the interpretation of our results is not impacted by this additional inclusion, though agree that it belongs there for completeness. Finally, for our GeneCIS experiments, we have included information about our utilized backbone network, which follows the one utilized in the original GeneCIS paper.
>
> ---
>
> __8. Limitations:__
> The reviewer correctly points out that while our approach offers several crucial benefits such as being high-performant, training-free, flexible and scalable, interpretable and intervenable, there is no free lunch, and a certain trade-off is to be expected. We will provide additional information for each limitation noted by the reviewer individually.
>
> * _On inference cost & efficiency compared to previous works:_ Indeed, while existing methods also utilize large VLMs, the use of distinct captioning and reasoning modules elicits additional runtime. On our hardware setup (V100 / A100 GPUs) when compared to trained methods, we find inference times to increase by ~1s. However, the modularity also allows our setup to directly benefit from efficiency changes within each module (as e.g. seen in the cost & inference time reductions for LLMs over the past year), allowing one system to remain operational for a long period of time. This stands in particular contrasts to other ZS-CIR methods that require costly training to incorporate any pipeline changes or shifts in expected datasets (e.g. Pic2Word trains the mapping network on 8 GPUs on the complete CC3M dataset).

---

> > ### Author Response · Authors · 2023-11-14
> > **Response to Review - continued**
> >
> > ---
> >
> > __Continuation for 8. Limitations__:
> > * _Error propagation through module-specific errors (also Q1):_ By combining different modules, error propagation can indeed happen - an incorrect caption results in a faulty modified retrieval caption, and incorrect reasoning can turn a sufficient caption into a wrong modified version. We do not claim our approach to be void of errors, however unlike existing works, our modular pipeline makes these actually much more transparent, and allows us to separate and reason by various error types (compared to black-box errors in previous approaches). We provide several studies for this in our paper, where we first separate failure cases in Fig. 3 and 4.3. And indeed, an incorrect caption affects the final reasoning, and incorrect reasoning affects an otherwise sufficient caption. However, the interpretability and modularity clearly separates these error sources. Indeed, our  intervention examples in Fig. 4 show how errors within each domain can be easily corrected individually to correct the complete retrieval process. Finally, beyond actual interpretability of errors and the possibility to intervene on them, our overall performance on most benchmarks is much stronger than previous trained methods, resulting in an overall smaller number of total errors.
> >
> > * _Overall compatibility of different modules._ We tested 6 VLMs, 4 LLMs and 3 captioning models. Controlling for each module change (ablation over captioning & reasoning modules in Tab. 3, scaling laws in Fig. 6), we found all modules to operate well together, with the main drop in performance stemming from shortcomings between module candidates.
> >
> > ---
> >
> > __9. Writing__
> > We have included all suggestions made by the reviewer across previously mentioned points. If there is anything else the reviewer would like to see addressed on the writing end, we will gladly incorporate that as well.
> >
> > ---
> >
> > __Q2. Supervised CIR__
> > The main goal of our approach (as well as other ZS-CIR methods) is to have a general purpose model that performs well across multiple benchmarks (i.e without specializing to a single setting) in the absence of expensive curated data. Most supervised CIR methods instead focus on training separate models specific for each benchmark, thereby losing generality. We illustrate this in the Appendix with the Combiner architecture, where we see that our approach nearly matches the performance of specific Combiner models on all of the benchmarks. Still, even the fully supervised CIR approaches have lackluster performance, nearly matched by our general purpose approach (even without scaling respective modules).
> >
> > This is partly due to the in parts ambiguous nature of the task (see e.g. Fig. 3 for an example), and consequently the lack of benchmarks that can account and evaluate for this ambiguity (for an image and a corresponding instruction, there are multiple suitable candidates - but these are generally not fully covered in existing benchmarks). However, as the strong performance of our training-free zero-shot approach showcases, it is also because of currently insufficient understanding on how to build more effective (ZS-)CIR systems.
> >
> > ---
> >
> > __Q3. CIRR evaluation details:__
> > The CIRR evaluation protocol has 2 settings. The first is where the target image has to be retrieved from the entire gallery. The second setting is the subset evaluation, where there are 6 handcrafted hard-negatives from which the correct target image must be chosen. We will highlight this information in the final draft.
> >
> > ---
> >
> > __Q4. Alternate Prompts:__
> > We test various alternative prompts, but find performance to be robust as long as the particular task remains sufficiently well described within it - this means that changes in general formulation have limited impact. Further gains can be achieved by extending the task description with general task examples, as has been shown in LLM prompting.

---

> ### Author Response · Authors · 2023-11-22
>
> With the discussion period coming to an end, we would like to thank the reviewer again for their helpful feedback, and hope that our replies, alongside strong additional experimental results and an updated draft have addressed all questions. Of course, we are happy to continue the discussion in case of any further concerns.

---

> > ### Comment · Reviewer_JtMi · 2023-11-23
> > **Official Comment by Reviewer JtMi**
> >
> > Thanks a lot for the authors' detailed responses. This work has its own sparkle (detailed in the comments from reviewers), while as said in the initial statement, many things remain to be done, including but not limited to polishing writing and providing more in-depth discussion (as the authors response). Therefore, I keep my rating.

---

> ### Author Response · Authors · 2023-11-23
>
> We appreciate the response by the reviewer, and thank them for appreciating our work.
>
> We do have extensively discussed issues raised by the reviewer, and have uploaded an updated draft, which has polished the writing based on the reviewers suggestions, while also including comprehensive additional, very positive results on Fashion-IQ.
>
> We also would like to note that a discussion on limitations is also available (see response), where we showcase possible failure cases (alongside the option to address them through interventions) or have experiments that showcase compatibility of different modules.
>
> If there is any other discussion the reviewer would like to see included and they believe has not yet been addressed, we would gladly do so.

---

### Official Review · Reviewer_QaMw · 2023-11-02

**Soundness:** 2 fair
**Presentation:** 2 fair
**Contribution:** 1 poor
**Rating:** 6
**Confidence:** 4

**Summary:**

This paper aims to achieve training-free zero-shot compositional image retrieval by using off-the-shelf models pretrained on large-scale datasets. The authors use language as an abstraction layer for reasoning about visual content. Particularly, it uses off-the-shelf vision-language models like BLIP-2 or CoCa to generate a detailed description of the query image. Next, an LLM or GPT-like model combines the input image description (from BLIP-2) and an input textual query (e.g., "shows only one of them, which is bigger and is next to a trash can") to generate a caption for the desired target image. Finally, a vision language model like CLIP performs text-to-image retrieval using the generated caption from LLM and images from an image database. This modular approach ushers a few benefits like: (i) the resulting approach is training-free and being modular, we can flexibly plug-and-play individual components like replace the LLM with either GPT or Llama, replace the VLM with BLIP-2 or CoCa; (ii) the composition happens in the human interpretable language domain offering human interpretability; and (iii) it is possible for humans to intervene on the retrieval process to fix or post-hoc improvements in retrieval results.

**Strengths:**

While there are a lot of works on compositional image retrieval, previous works typically require training several components like textual inversion and lack human interpretability. The method proposed in this paper avoids all these problems by simply composing image and textual query in the language domain.

On the surface this is ingenious because you do it in a modular way and each module is a highly-generalisable large-scale pre-trained model. For example, when training a textual inversion, we do not really guarantee it scales to open-set setups (given we train them in limited data and compute). We just use a small-scale network and hope the rest of powerful models take care of it. The proposed method works zero-shot and you know it works for open-set setups.

Additionally, unlike prior trainable compositional image retrievals, there is a lot of effort that goes into interpretability -- "how was the image and query text was composed". Since the proposed method does this composition entirely in the language space, you know exactly what information was extracted from the image and you can see the generated caption from LLM for the desired target image. This not only makes the retrieval process highly transparent, but also allows post-hoc edits -- you can literally make changes to adjust your retrieval results.

I encourage more works that reuses as much as large-scale models and combines them as modules with minimal or training-free way.

**Weaknesses:**

Despite its appeal, there are a few important drawbacks. This entire process sticks on the underlying assumption that our image captioning module (e.g., BLIP-2 or CoCa) can provide a "detailed caption" that captures all information.

This, in my opinion, is a strong assumption. A lot depends on your captioning module. While the captioning module may be super accurate, it can miss some "less important" details that I want to change. For example, given a photo, the sky was orange and my textual query is "make the colour of the sky darker". If the image captioning module omits the colour of sky (i.e., orange) and focuses on the foreground (e.g., a person holding a flower, sitting in a bench near a park where kids are playing) -- the rest of the modules have no way of combining my textual query "make the colour of sky darker".

This is an example, where the entire pipeline fails -- due to no fault of the image captioning module. It provided a detailed image description, but it is not possible to describe every little "unimportant things".

Looking at the same problem from a different direction, is the information bandwidth of an image the same as language? (an image can be worth a thousand words)

**Questions:**

I am confused, how is the proposed method different from VisProg (https://arxiv.org/pdf/2211.11559.pdf)?
It feels I can get the same benefits of the proposed method by using the VisProg paper.

---

> ### Author Response · Authors · 2023-11-14
> **Response to Review**
>
> We thank the reviewer for their thoughtful comments. We appreciate that they found our paper ingenious, while highlighting the particular benefits with respect to interpretability and the capacity to allow for interventions. We address the concerns below:
>
> ---
>
> __1. Is a caption sufficiently descriptive?__
> We agree that a simple textual caption is limited in its ability to fully reflect the content of an image. This is something we also specifically highlight throughout the paper, particularly when addressing failure cases (Fig. 3) and their addressal through interventions in Fig. 4.
>
> Generally however, when compared to current methods for Zero-shot Compositional Image Retrieval (ZS-CIR) which focus on mapping an input image to a single token in the textual embedding space, CIReVL is far less restrictive - generating a 10-20 word caption which can leverage e.g. 50+ tokens to represent the information in the image.
>
> In addition, operating in the language space allows our approach to be inherently interpretable, and actually intervenable.
> Given these benefits, our work particularly tries to emphasize how far one may push a language-centric compositional image retrieval approach, and successfully shows that compared to existing methods, current benchmarks can be effectively tackled by purely representing each image with a caption/description.
>
> Future work should focus on developing benchmarks that have information that cannot be easily described with a single description. In the current benchmarks, these cases are not the majority, while we do illustrate a potential example in Fig. 3.
>
> ---
>
> __2. Differences to Visual Programming:__
> While related, there are several significant differences between our proposed approach and Visual Programming.
> Visual Programming aims to solve a variety of vision and language tasks at the same time using an LLM to control the APIs to various modules.
>
> However, all the tasks tackled by Visual Programming typically relate to working with a single image at a time, and don’t account for large-scale applications in which e.g. retrieval over an expansive gallery set has to be performed. To possibly adapt the Visual Programming method to this task, one would need to precompute the image features for all the images in the gallery set, provide sufficient in-context examples and a task description to enable the LLM to reason over the description, and provide particular instructions on how to find the correct final image. All of these modifications involve significant specialization to the CIR task and move the end result far away from the Visual Programming generalist.
>
> In general, VisProg is a general framework which leverages different specialist modules and figures out ways to recombine them. Consequently, CIReVL should be regarded more as an expert, interpretable compositional image retrieval module that can then be leveraged by generalist agents via e.g. VisProg.

---

> > ### Comment · Reviewer_QaMw · 2023-11-23
> >
> > > Generally however, when compared to current methods for Zero-shot Compositional Image Retrieval (ZS-CIR) which focus on mapping an input image to a single token in the textual embedding space, CIReVL is far less restrictive - generating a 10-20 word caption which can leverage e.g. 50+ tokens to represent the information in the image.
> >
> > I do not agree with this statement. (i) Textual inversion is an optimisation-based approach that (to some extent) guarantees the text token is "faithful" to the image with respect to the image retrieval task. The same cannot be said about the image captioning model that "looks" at an image and generates a 50-word caption. (ii) Also, comparing the textual feature (from inversion) with 50 words may not be a correct evaluation ("Interpreting the Learned Prompts" in https://arxiv.org/pdf/2109.01134.pdf)
> >
> > Regarding my comment:
> > ```While the captioning module may be super accurate, it can miss some "less important" details that I want to change.```
> > Fig-4 seems like a patchwork to me -- if a human has to modify the generated image caption, he/she might as well write it in the first place.
> >
> > Although not a definite solution, the authors can try:
> > 1. Use the input text to "prompt" BLIP2 ($\Psi_{C}$) to generate a "relevant" image caption. This prompt can be hand-engineered or learned.
> > 2. BLIP2 now generates a caption which is mindful of the input text -- hence, it does not miss the "less important details that I want to change".
> > 3. The pipeline stays interpretable -- as the image+text combination happens in the textual domain.
> >
> > I am still keeping my rating as weak reject (score 5), but the ACs may feel to accept the paper nonetheless.

---

> ### Author Response · Authors · 2023-11-22
>
> With the discussion period coming to an end, we would like to thank the reviewer again for their helpful feedback, and hope that our replies, alongside strong additional experimental results and an updated draft have addressed all questions. Of course, we are happy to continue the discussion in case of any further concerns.

---

> ### Author Response · Authors · 2023-11-23
>
> We thank the reviewer for their detailed reply and reference, and would like to note two things:
>
> ---
>
> >  Textual inversion is an optimisation-based approach that (to some extent) guarantees the text token is "faithful" to the image with respect to the image retrieval task. The same cannot be said about the image captioning model that "looks" at an image and generates a 50-word caption.
>
> If e.g. looking at SEARLE, the textual inversion network is trained to generate a token that simply mimics classnames/concepts - so information available through the generated token will still be very lackluster.
> In addition to that, the computed text token is then incorporated into a full input prompt which is then fed into a text embedding system - and there is only so much information that can be packed into a single text token to account for the entire CIR task.
>
> Taken together, a lot of relevant context will still go missing - and without a clear mechanism for interpretability, it is unclear what exactly.
>
> On the other hand, there is no limit to the captions we utilize, and as the reviewer rightly highlighted, the captioning system can be extended arbitrarily (if needed), without impacting the overall pipeline.
> The generated captions generally provide much more context (as shown also qualitatively in the paper) beyond a single concept, and the resulting high performance gains we achieve - even when using the same retrieval system - support this.
>
> ---
>
> > While the captioning module may be super accurate, it can miss some "less important" details that I want to change. Fig-4 seems like a patchwork to me -- if a human has to modify the generated image caption, he/she might as well write it in the first place.
>
> This is of course true - the captioning system is not perfect, but given the modular nature, can simply we replaced in a plug-and-play fashion if needed. However, we show that even with very general open-source captioning models, that very strong CIR performance can be achieved.
>
> Regarding the human just writing the caption themselves, this is exactly the point of the intervenability, and a byproduct of our inherent interpretability.
> If the human believes that insufficient context is extracted from an image, and wants to guarantee a more faithful retrieval, they can also write the caption themselves.
> Such an intervention into the retrieval process is not possible for other trained systems, and is simply a bonus point on top of the modular and interpretable CIR framework.

---

### Official Review · Reviewer_W9gL · 2023-11-03

**Soundness:** 3 good
**Presentation:** 3 good
**Contribution:** 2 fair
**Rating:** 6
**Confidence:** 4

**Summary:**

The paper proposes a training-free solution for addressing the compositional image retrieval problem. The idea is quite simple: i) take the pre-trained VLM model to describe what the input image contains, ii) use an LLM to combine the description/caption of the input image with the desired modifications as described by the user query, finally, iii) use the resulting text to generate VLM (i.e., CLIP) representations and search the target image database via knn search. The solution is evaluated against multiple baselines (PALAVRA, Pic2Word, SEARLE) and datasets (CIRCO, CIRR, GeneCIS) and provides better results.

**Strengths:**

- The paper is technically sound and widely applicable not only for text-image use case, but also text-video and probably other problem domains.
- The paper investigates three different datasets against multiple baselines and various additional ablations make the paper and results more appealing. Limitations of the proposed method also gets discussed with examples.
- The paper is easy to follow / well-written.

**Weaknesses:**

- [Novelty & Literature review] There are a wider number of papers in the field since last year and some of those papers requires mention (at least the ones before the ICLR submission deadline). The most similar paper ((https://arxiv.org/pdf/2310.05473.pdf) proposes the same approach but with an additional training method for merging the automatically generated image caption and the desired modification's text description. This paper contains additional papers/baselines to cite/include. It would be great if there is a way to understand whether the proposed training-free method would perform better or worse compared to the paper mentioned above.

- [Experiments] Baseline PALAVRA attack different set of problems, uses different datasets and metrics. It does not mention CIR task in the text. Not clear if PALAVRA serves as a baseline.

**Questions:**

- A short discussion about extensibility / applicability of the proposed approach could be beneficial for the research community.
- FashionIQ is yet another dataset which other papers use for CIR evaluations, it might be worth considering it in the future.
- Is there a way to detect when the LLM fails to provide a meaningful / valid re-composed text ? How much room left to improve this step? What is the future research direction?

---

> ### Author Response · Authors · 2023-11-14
> **Response to Review**
>
> We thank the reviewer for their thoughtful comments. We appreciate that they found our paper technically sound, widely applicable, easy to follow and with rigorous experimentation.
> We address the concerns below:
>
> ---
>
> __1. Novelty of the proposed approach:__
> We thank the reviewer for pointing out the additional reference. However, the paper is contemporary to our submission (i.e. it was published on arxiv after the ICLR deadline), thus we could not have provided a comparison with it/known about its existence. We acknowledge several related works in Section 2 of our main paper, and highlight crucial differences between these works and our approach.
>
> Most works on compositional image retrieval aim to train/fine-tune models specific for each dataset (i.e., training separate models for CIRR, Fashion-IQ etc.). Zero-shot Compositional Image Retrieval (ZS-CIR) instead aims to train a single model that generalizes across multiple benchmarks without the use of paired (source image, text, target image) data. The only works that explicitly focus on this task (and that we compare with) are Pic2Word [Saito et al 2023] and SEARLE [Baldrati et al. 2023], which are based on a different principle, i.e. trained textual inversion. Both methods already leverage off-the-shelf, large-scale pre-trained models as part of their pipeline.
>
> In our work, we go one step further and propose the first training-free approach for this task, that only relies on off-the-shelf pre-trained models, while allowing for easier scalability, interpretability and the possibility of interventions.
>
> ---
>
> __2. PALAVRA as a baseline:__
> We agree that PALAVRA has not been proposed for the CIR task. However, since it is conceptually very similar to Pic2Word and SEARLE, it was originally introduced as a comparison in these works. To retain direct, comprehensive comparability, we also utilize PALAVRA as a corresponding baseline.
>
> ---
>
> __3. Results on Fashion-IQ:__
> We thank the reviewer for the suggestion. We have now updated our paper with the results on the Fashion-IQ benchmark. For the detailed results, we refer to our shared comment. In summary, we find that CIReVL significantly outperforms existing methods on all three subtasks, e.g. improving average Recall@10 by nearly 6 percentage points (pp) (28.29% versus 22.89% for SEARLE) with a ViT B/32 backbone. In addition, the simple scalability allows us to hot-swap the retrieval model to achieve additional 4pp (increasing from 28.29% to 32.19%).
>
> ---
>
> __4. Short Discussion on extensions and applications of our approach:__
> The reviewer correctly points out that our method would in-principle be applicable to videos as well as other forms of retrieval contexts, such as with interleaved images and texts (e.g. visual storytelling).
>
> While our exact setup is structured to operate and scale especially well to the compositional image retrieval task, as long as the particularities of the problem at hand can be reasoned over in the language domain while retaining a downstream compositional retrieval or alignment target, CIReVL can certainly be adapted to account for this.
>
> We believe this to be of definite interest for future research, and will expand our discussion to incorporate the arguments above.
>
> ---
>
> __5. Detect LLM Failures and future steps:__
> Indeed, it is not easy to automatically detect failures in the captioning model or the LLM reasoning, even though our intervention examples highlight a clear need to account for these shortcomings, and the benefits of addressing them.
>
> For instance, one strategy may be in using another LLM to verify if the generated description is plausible or not, given the initial caption and instruction. However, doing so can only tackle issues in the LLM reasoning part, and is not able to address failures in the image captioning.
>
> As can be seen, finding an automated mechanism to detect all manners of intervention cases is a difficult problem on its own, albeit a crucial component to improve the performance of compositional image retrieval systems. It is something we would love to see tackled in future work.

---

> ### Author Response · Authors · 2023-11-22
>
> With the discussion period coming to an end, we would like to thank the reviewer again for their helpful feedback, and hope that our replies, alongside strong additional experimental results and an updated draft have addressed all questions. Of course, we are happy to continue the discussion in case of any further concerns.

---

> > ### Comment · Reviewer_W9gL · 2023-11-22
> >
> > Although other reviewers expressed concerns over novelty, the paper proposes a simple (easy to re-implement) baseline for future work in this research area. The solution also provides outstanding results compared to other ZS-CIR baselines. I am in favor of increasing the initial rating. The authors also largely addressed my previous questions. Additional experiments on Fashion-IQ, discussions on novelty and future work are also convincing. This also contributed to the final decision.

---

### Author Response · Authors · 2023-11-14
**Shared Reply**

We thank the reviewers for taking the time to provide detailed and insightful reviews!
In particular, we appreciate the reviewers acknowledging the excellent presentation / writing (Tg9t,W9gL) & clear visualizations (JtMi) in our paper, our thorough experimentation (JtMi), and the intuitive, technically sound structure of our CIReVL model (Tg9t,W9gL) alongside its promising results (Tg9t), wide applicability (W9gL) and ingenious modular structure (QaMw), which allows it to be interpretable (QaMw,Tg9t) and intervenable (QaMw,JtMi,Tg9t) unlike previous works.

We have addressed the specific comments with individual replies, and will include changes to the main document shortly. We also summarize our additional experiments on the Fashion-IQ benchmark.

---

__Fashion-IQ Benchmark.__ We would like to thank the reviewers (W9gL,Tg9t) for pointing out Fashion-IQ as a valuable testbed for the CIR task. Running CIReVL on all three Fashion-IQ subtasks, we find significant gains over comparable methods, raising e.g. the average Recall@10 by over 5pp (R@50 over 6pp). Even more, the natural modularity allows us to simply change the retrieval module for additional 4pp gains. Our results here are extremely positive and we will add this to our main paper. We believe that this suggestion of the reviewers has strengthened our paper significantly.

---

Backbone: __ViT-B/32__

| Setup | Shirt R@10 | R@50 | Dress R@10 | R@50 | Top R@10 | R@50 | Avg R@10 | R@50 |
|---|---|---|---|---|---|---|---|---|
| Text-Only| 19.87 | 34.99 | 15.42 | 35.05 | 20.81 | 40.49 | 18.70 | 36.84 |
| Image-Only| 6.92 | 14.23 | 4.46 | 12.19 | 6.32 | 13.77 | 5.90 | 13.37 |
| Image+Text | 13.44 | 26.25 | 13.83 | 30.88 | 17.08 | 31.67 | 14.78 | 29.60 |
| PALAVRA | 21.49 | 37.05 | 17.25 | 35.94 | 20.55 | 38.76 | 19.76 | 37.25 |
| SEARLE   | 24.44 | 41.61 | 18.54 | 39.51 | 25.70 | 46.46 | 22.89 | 42.53 |
| CIReVL    | 28.36 | 47.84 | 25.29 | 46.36 | 31.21 | 53.85 | 28.29 | 49.35 |

Backbone: __ViT-L/14__

| Setup | Shirt R@10 | R@50 | Dress R@10 | R@50 | Top R@10 | R@50 | Avg R@10 | R@50 |
|---|---|---|---|---|---|---|---|---|
| Pic2Word | 26.20 | 43.60 | 20.00 | 40.20 | 27.90 | 47.40 | 24.70 | 43.70 |
| SEARLE  | 26.89 | 45.58 | 20.48 | 43.13 | 29.32 | 49.97 | 25.56 | 46.23 |
| CIReVL   | 29.49 | 47.40 | 24.79 | 44.76 | 31.36 | 53.65 | 28.55 | 48.57 |

Backbone: __ViT-G/14__
| Setup | Shirt R@10 | R@50 | Dress R@10 | R@50 | Top R@10 | R@50 | Avg R@10 | R@50 |
|---|---|---|---|---|---|---|---|---|
| CIReVL   | 33.71 | 51.42 | 27.07 | 49.53 | 35.80 | 56.14 | 32.19 | 52.36 |

---

### Meta-Review · Area_Chair_SCpY · 2023-12-05

**Metareview:**

This work addresses the problem of compositional image retrieval by combining a generated image caption with the textual context, using an LLM, and then generating an image embedding from the combined text that can be used to search a database. The paper received diverging ratings which remain after the discussion periods. All reviewers agree that the novelty is somewhat marginal, but the approach is sound/intuitive. There were also concerns initially about the experimental validation e.g. limited domains, missing ablations, and inappropriate baseline. During the discussion period, the authors included significant new experimental results on Fashion-IQ which greatly reduced these concerns. The AC recommends acceptance as the paper proposes a simple method that gives significant performance gains for CIR.

**Justification For Why Not Higher Score:**

The novelty is marginal

**Justification For Why Not Lower Score:**

The experimental results are strong.

---

### Decision · Program_Chairs · 2024-01-16

Accept (poster)